# Can spatial filtering separate voluntary and involuntary components in children with dyskinetic cerebral palsy?

**Cassie N. Borish**[1], **Matteo Bertucco**[2]*, **Denise J. Berger**[3,4], **Andrea d'Avella**[3,5], **Terence D. Sanger**[6,7,8]

**1** Department of Biomedical Engineering, University of Southern California, Los Angeles, California, United States of America, **2** Department of Neurosciences, Biomedicine and Movement Sciences, University of Verona, Verona, Italy, **3** Laboratory of Neuromotor Physiology, Foundation Santa Lucia, Rome, Italy, **4** Department of Systems Medicine and Centre of Space Bio-medicine, University of Rome Tor Vergata, Rome, Italy, **5** Department of Biomedical, Dental, Morphological and Functional Imaging Sciences, University of Messina, Messina, Italy, **6** School of Engineering, University of California, Irvine, California, United States of America, **7** School of Medicine, University of California, Irvine, California, United States of America, **8** Children's Hospital of Orange County, Orange, California, United States of America

☯ These authors contributed equally to this work.
* matteo.bertucco@univr.it

**Data Availability Statement:** All relevant data are within the paper and its Supporting information files.

## Abstract

The design of myocontrolled devices faces particular challenges in children with dyskinetic cerebral palsy because the electromyographic signal for control contains both voluntary and involuntary components. We hypothesized that voluntary and involuntary components of movements would be uncorrelated and thus detectable as different synergistic patterns of muscle activity, and that removal of the involuntary components would improve online EMG-based control. Therefore, we performed a synergy-based decomposition of EMG-guided movements, and evaluated which components were most controllable using a Fitts' Law task. Similarly, we also tested which muscles were most controllable. We then tested whether removing the uncontrollable components or muscles improved overall function in terms of movement time, success rate, and throughput. We found that removal of less controllable components or muscles did not improve EMG control performance, and in many cases worsened performance. These results suggest that abnormal movement in dyskinetic CP is consistent with a pervasive distortion of voluntary movement rather than a superposition of separable voluntary and involuntary components of movement.

## Introduction

Children with tetraplegic or dyskinetic cerebral palsy (CP) suffer from movement disorders such as muscle weakness, spasticity, dystonia, and dyspraxia that can prevent meaningful voluntary movement [1]. As a result, these children have a very limited ability to do things that typically developing children do, including playing, interacting spontaneously, exploring, and using movement for communication. For such children, assistive devices may provide

**Funding:** This research was supported by Department of Biomedical Engineering, University of Southern California, Los Angeles, CA, USA.

mobility, and in more extreme cases, functional communication. However, careful consideration must be placed into the interface of such devices. Children may be able to use low-bandwidth interfaces such as a head switch, force-control joystick, or button interface, however, their output from these devices is often slow and limited by their own movements. In a previous study, we found that children with CP who depend on a touch-screen interface to communicate generate an average of only 50 words per week [2]. In essence, children can no better control such devices than they can their own body because the interface is limited by the child's own difficulties with movement. Therefore, if we are to provide assistive devices for this group of children, we must address the problem of optimal extraction of voluntary controllable signals from involuntary, unwanted muscle activity. Ultimately, we want to provide assistive devices that will allow these children to explore and manipulate their environment in ways that typically developing children do. Children need flexible real-world interfaces whose motions are not described in advance, so that they can learn and develop their own movements, and explore varying and unpredictable goals.

The major barrier for children with dyskinetic CP is not the design of the device to be controlled, but the interface that the child uses to control the device. Myocontrol, the control of devices using electromyographic (EMG) signals, may be used to allow children with dyskinetic CP to control assistive devices in a flexible manner. Myoelectric signals have been used for biofeedback and functional electrical stimulation for rehabilitation [3–9], as well as the control of other external devices such as exoskeletons and speech synthesizers [10,11]. Unlike in spinal cord injury, in CP there is no disconnect between the brain and the spinal cord. Therefore, the EMG signal provides a direct read-out of movement-related activity in motor cortex. Myocontrol is preferable to brain-computer interfaces, which are either invasive (requiring implantation in the brain) or low bandwidth (when using scalp electrodes). It does not restrict where the child can look, as eye gaze control would. It also allows for smooth and flexible control, as opposed to button or other on/off interfaces. The challenge with myocontrol for children with dyskinetic CP is that the EMG signal, like the child's movements, is not completely voluntarily controllable. It would therefore be helpful to separate the voluntary component of the EMG signal from the involuntary component in order to control a device.

A previous study hypothesized that abnormal movements in dyskinetic CP may be due to an inability to suppress unwanted components of movement [12]. Studies of reaching and other upper extremity movements in children with CP have found increased variability in their movements, substantiating this hypothesis [12–15]. However, the origin of the variability is still unknown. The noise might reflect unrelated neural activity. It could also be the result of a "noise generator" injecting a new source of noise. The ability to characterize the noise in dyskinetic CP could provide insight into the nature of the movement abnormality and inform the development of future treatments.

Bernstein proposed that the control of multiple muscles could be simplified by selecting a small set of patterns, or synergies, to reduce the high dimensionality of the set of possible actions [16]. It has since been shown that kinematics and EMG patterns in humans occupy low dimensional spaces for specific tasks [17–20]. Synergies likely play a role in aiding typically developing participants to learn and accomplish new skills [21]. However, synergies in children with dystonia due to CP are likely to be distorted by co-contraction [8,9], signal-dependent noise [14,22–24], and weakness [25]. Thus, selecting the most controllable synergies is particularly important and could provide a method for separating controllable from uncontrollable components of movement for children with CP. Extending Bernstein's hypothesis from normal to abnormal movements, we anticipate that, like normal movement, dyskinetic movement can be characterized by a superposition of synergies. However, unlike normal

movement, it is possible that in CP some of the synergies are not under voluntary control and appear instead to be modulated by noise.

Our methods are based on two assumptions: (1) noise in CP is low-dimensional and (2) the dimension of noise can be identified and isolated by nonnegative matrix factorization (NMF) [26]. These assumptions are based on previous work that used NMF to reduce noise in speech enhancement and force reaching task in a virtual environment with healthy individuals [27,28]. By observing how myocontrol performance changes using a reduced EMG representation, we could potentially learn how noise exhibits itself in CP. If the noise falls primarily on one component and that component is removed, myocontrol performance might improve. However, if noise is inherent in all neuronal channels, components would contain a mix of controllable and noisy signals. Thus, removing a component would not improve myocontrol performance. For this study, we performed a synergy-based decomposition on eight muscles of the upper limb. We used a one-dimensional Fitts' Law task to determine which components were most controllable. Similarly, we determined the most controllable individual muscles. We then compared whether removing the least controllable components or muscles improved myocontrol performance in a two-dimensional task.

Fitts' Law is a well-established model for the trade-off between speed and accuracy [29], who showed that the movement time (MT) to achieve a target in proportional with increased movement distance (D) and decreased target size (W). The Fitts' Law was derived from the information theory by Shannon [30] and adapted to the human motor system. Given its direct linkage with information theory and the channel capacity theorem, Fitts's Law has effectively applied in the field of human-computer interfaces (HCI) as a tool to maximize the information rate for a given device, such as mice and touchscreens [31,32]. Therefore, Fitts's law proves to be also a valuable tool to assess controllability of myocontrol interfaces [33–37].

This study tested whether voluntary and involuntary components of movements in children with cerebral palsy could be linearly separated. We used Fitts' Law to assess the controllability of the components, identified by synergy decomposition, and individual muscles, and tested whether removing less controllable components improved myocontrol performance. In contrast to our initial expectation, removal of less controllable components did not improve performance, suggesting that in dyskinetic CP loss of controllability affects all components of movement.

## Materials and methods

### Participants

Inclusion criteria for this study were: (I) dyskinetic CP with dystonia affecting at least one upper extremity; (II) pediatric age (7–21 years); (III) cognitive ability sufficient for understanding task instructions. Five children with cerebral palsy (4 males, 1 female; ages 11 to 18 years, mean 15 ± 3 years) performed this protocol (Table 1). The University of Southern California Institutional Review Board approved the study protocol (UP-12-00457). All parents gave informed written consent for participation and all children gave written assent. Authorization for use of protected health information was signed in accordance with the Health Information Portability and Accountability Act. The study was performed in accordance with the Declaration of Helsinki.

### Experimental setup

The set-up was similar as used in previous studies [27,38]. Participants sat in front of a desktop with their more affected forearm inserted in a splint below the desktop immobilizing the hand, wrist, and forearm with respect to the desktop. The splint is connected by a steel bar to a 6-axis

**Table 1. Clinical characteristics of subjects with dyskinetic cerebral palsy.**

| ID | Sex | Age | BAD Scale Score | | Arm Tested | Diagnosis | Symptoms |
|----|-----|-----|-----|-----|-----|-----|-----|
| | | | L Arm | R Arm | | | |
| 1 | M | 14 | 1 | 2 | R | Cerebral palsy, 7 weeks premature | Upper-limb dystonia and bilateral lower-limb spasticity |
| 2 | M | 18 | 1 | 0 | L | Cerebral palsy | Left arm dystonia |
| 3 | F | 16 | 2 | 3 | R | Hypoxic ischemic injury | Generalized dystonia |
| 4 | M | 14 | 2 | 0 | L | Hypoxic ischemic injury, secondary to stroke at birth | Left arm dystonia |
| 5 | M | 11 | 3 | 3 | L | Hypoxic ischemic injury | Generalized dystonia |

Abbreviations: BAD Scale, Barry-Albright Dystonia Scale; F, female; L Arm, severity of left arm; M, male; R Arm, severity of right arm. Scores are based on the BAD Scale; for each segment, the score ranges from 0—absence of dystonia to 4—severe dystonia.

force transducer (Delta F/T Sensor, ATI Industrial Automation, Apex, NC, USA) rigidly attached below the table to record isometric forces and torques. The center of the palm was aligned with the body midline at the height of the sternum, and the elbow was flexed by approximately 90˚. The participants' view of their hand was occluded by the desktop.

Feedback was provided by a 21-inch LCD monitor inclined with its surface perpendicular to the participants' line of sight approximately 1 m away (Fig 1). After calibration, the monitor displays a circular cursor indicating the force magnitude and direction at the hand as well as a set of 8 circular targets. At rest, the cursor lies in the center of the screen, with the 8 targets arranged in a circle to create a "center-out" virtual reaching task. Subjects attempt to move the cursor to one of the targets by applying isometric force to the wrist splint in the appropriate direction with appropriate magnitude. Rotational torque at the splint was ignored.

Surface electromyographic (EMG) activity was recorded from the following eight muscles: pectoralis major (PEC), brachioradialis (BRACH), biceps (BIC), triceps (TRIC), anterior deltoid (AD), lateral deltoid (LD), posterior deltoid (PD), and middle trapezius (TRAP). EMG activity was recorded with bipolar electrodes (DE–2.1, Delsys Inc., Boston, MA, USA), band-pass filtered (20–450 Hz) and amplified (gain 1000, Bagnoli–8, Delsys Inc.). Force and EMG data were sampled at 1 KHz using an analog-to-digital interface (Power 1401, CED Technologies Inc., UK) and custom data acquisition software. The force data were low-pass filtered ($2^{nd}$ order Butterworth, 1 Hz cutoff), while a non-linear Bayesian filter ($\alpha = 1e-4$, $\beta = 1e-18$, 128-bin histogram) was applied to the rectified EMG signals [39]. Cursor position was proportional to either the actual force recorded by the transducer (force-control), or the force estimated in real-time from the recorded and rectified EMGs or synergies [27].

## Experimental protocol

Participants were tested on three separate days. On each day, they performed a two-dimensional (2-D), isometric, goal-directed speed-accuracy center-out aiming task using myocontrol under one of three conditions: 1) All EMG control, 2) Select Muscle control, and 3) Select Synergy control. "All EMG" control refers to the estimation of force from all recorded EMGs. "Select Muscle control" is the estimation of force from the most controllable muscles. "Select Synergy" control is the estimation of force from the most controllable synergies.

On each day, participants initially performed two blocks of trials in force-control. In the first force-control block, the mean maximum voluntary force (MVF) along eight directions (separated by 45˚) (Fig 1) was estimated as the mean of the maximum force magnitude recorded across 16 successful trials (two for each direction) in which participants were instructed to generate maximum force in each direction.

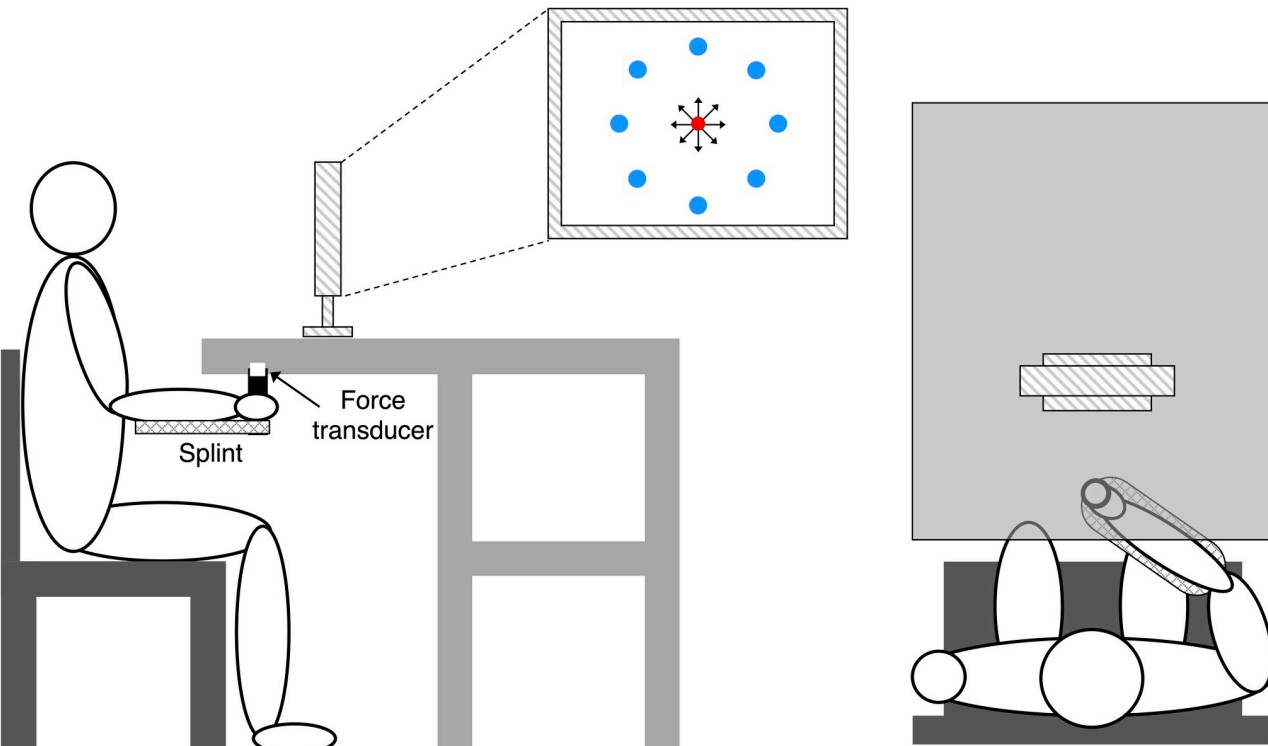

**Fig 1. Experimental setup.** Participants sat in front of a desktop with their more affected forearm inserted in a splint immobilizing the hand, wrist, and forearm. The center of the palm was aligned with the body midline at the height of the sternum, and the elbow was flexed by approximately 90˚. The participants' view of their hand was occluded by a table wherein an LCD monitor was positioned about 1 m away. A steel bar at the base of the splint was attached to a force transducer positioned below the table to record isometric forces. Subjects were instructed to perform a goal-directed speed-accuracy aiming task in which they had to maintain the cursor in a central start location (red circle ●) for 1 s, reach a target as soon as it appeared at one of 8 peripherals targets (blue circles ●), and maintain the cursor at the target for 0.2 s.

Participants were instructed to move the cursor quickly and accurately from the rest position to a target in one of the eight directions by applying forces on the splint. At the beginning of each trial, participants were asked to maintain the cursor within a red circle at the central start position for 1 s (tolerance of 2% MVF). Next, a "go" signal was given by displaying a blue circular target while the start red circle disappeared (Fig 1). Participants were instructed to reach the target as quickly as possible and to remain there for at least 0.2 s (tolerance of 2% MVF). Trials in which participants successfully stabilized in the target for 0.2 seconds were considered successful. Movement time (MT) was calculated as the time from when the cursor velocity was greater than 5% peak velocity during the trial ($t_0$) to the time once they successfully stabilized in the target for 0.2 seconds. For unsuccessful trials, MT was set at 7.5 seconds, which was the maximum trial length. If an unsuccessful trial occurred, the participant immediately repeated the trial. The limit set at 7.5 seconds was defined based on pilot data testing and allowed to provide enough time to complete the trial by considering the possibility that involuntary sustained or intermittent muscle contractions due to dystonia could occur during the performance.

In the second force-control block, participants performed 24 trials to targets positioned at force magnitudes corresponding to 10, 20, and 30% MVF (random order within cycles of eight directions) with a target width (diameter) corresponding to 5% MVF, presented in random order. The cursor position was linearly proportional to the EMG such that 0% MVF was the midpoint of the screen and 50% MVF was the edge of the screen. The EMG-to-Force mapping

was calculated from the data collected during this block. After this block, recorded data were processed to construct the myoelectric controller.

For the Select Muscle and Select Synergy conditions, the most controllable muscles and synergies were selected as detailed in Muscle and Synergy Selection section. Participants then performed the 2-D task, using myocontrol to reach targets in eight directions (separated by 45˚), three indexes of difficulty (ID, see below) per direction for 24 targets total. Targets corresponded to a distance of 15% MVF from the origin with widths 4.31, 7.24, and 12.18% MVF, resulting in IDs equal to 2.8, 2.05 and 1.3 bits respectively, and were presented in random order. The Fitts' Index of Difficulty (ID) was calculated with the Fitts' formulation [29]:

$$ID = log_2\left(\frac{2D}{W}\right) \qquad (1)$$

where D is the distance from the start position to the center of the target and W is the width of the target [29,31]. For Select Muscle and Select Synergy conditions the cursor position was linearly proportional to the EMG such that 0% MVF was the midpoint of the screen and 25% MVF was the edge of the screen. Participants performed 240 trials for each condition. Between trials, participants could rest for at least 15 seconds in order to reduce possible effects of fatigue.

## EMG-to-force mapping

Under isometric conditions, the force generated at the hand is approximately a linear function of the activation of muscles:

$$f = Hm + e_f \qquad (2)$$

where f is the generated 2-dimensional force vector, m is the 8-dimensional vector of muscle activations, H is a matrix relating muscle activation to force (dimensions 2 x 8), and $e_f$ is a 2-dimensional vector of force residuals [27] (Fig 2). Each column of H is the pulling direction of one muscle in the 2D plane (Fig 2, Row A). The EMG-to-force matrix (H) was estimated using multiple linear regressions of each applied force component with EMG signals recorded during the second force-control block. Only data from the dynamic phase of trials were used for regression, from $t_0$ until the cursor was stabilized in the target for 0.2 seconds. The force components were normalized to MVF and the maximum EMG activity during the generation of MVF in the first force-control block respectively.

## Synergy extraction

Muscle synergies were identified by Non-negative Matrix Factorization (NMF) from EMG patterns during the force control block using data from $t_0$ until the participant stabilized the cursor within the target for 0.2 seconds (dynamic phase). Synergy extraction is described by:

$$m = Wc + e_m \qquad (3)$$

where W is an $M$ x $N$ synergy matrix whose columns are vectors specifying relative muscle activation levels ($N$ number of synergies, and $M$ number of muscles), c is an $N$-dimensional synergy activation vector, and $e_m$ is an M-dimensional vector of muscle activation residuals [26,27]. NMF was implemented using the multiplicative update rule and the algorithm stopped when the reconstruction error ($R^2$) was not increased more than $10^{-4}$ for 10 consecutive iterations, or when a maximum number of $10^5$ iterations was reached. Synergy extraction was repeated with the number of synergies ($N$) ranging from 5 to 8 [27,38].

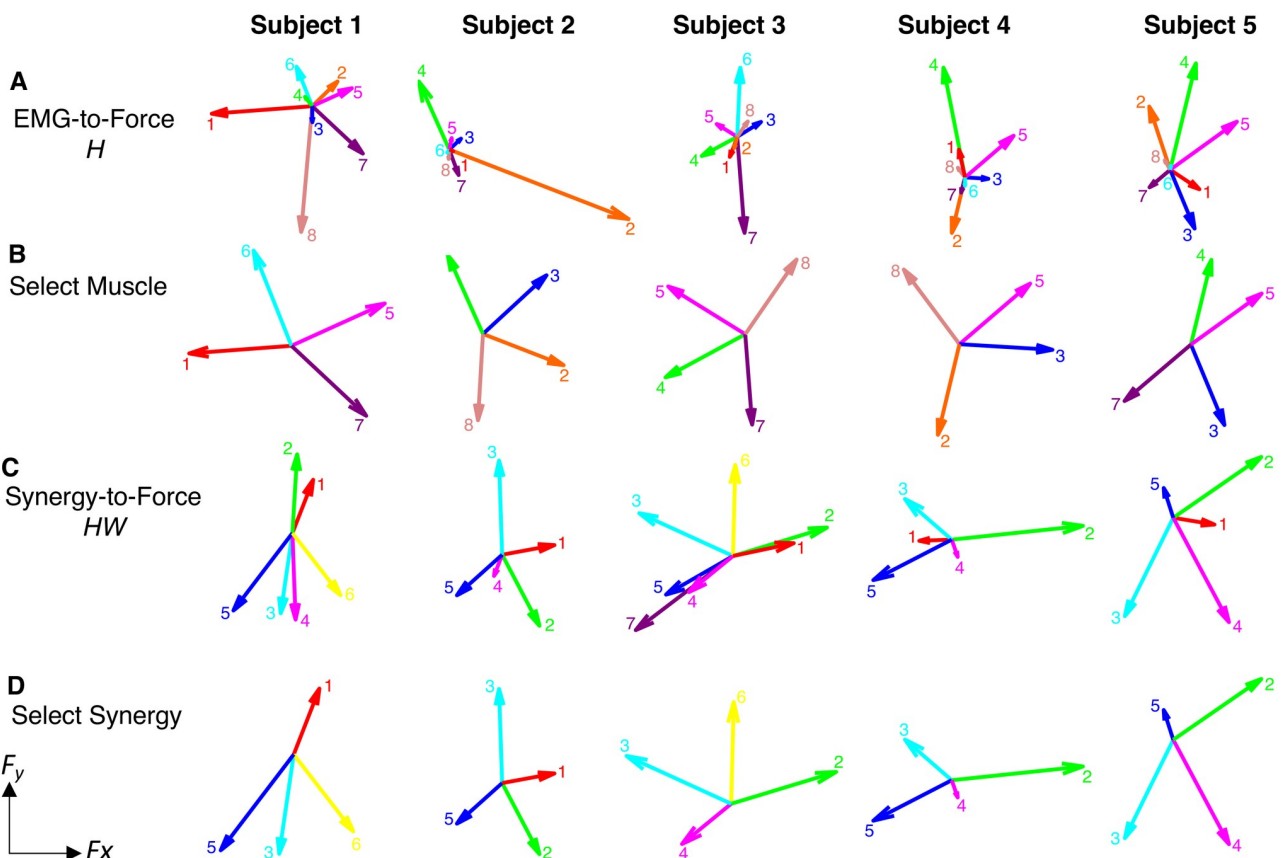

**Fig 2. EMG-to-force mapping and synergies.** Row A) EMG-to-force matrix H estimated for each participant from EMG and force data recorded during the force-control block. Each arrow represents the planar force generated by one muscle (1: pectoralis major (PEC), 2: brachioradialis (BRACH), 3: biceps (BIC), 4: triceps (TRIC), 5: anterior deltoid (AD), 6: lateral deltoid (LD), 7: posterior deltoid (PD), 8: middle trapezius (TRAP); Row B) The reduced mapping for the Select Muscle condition in the 2-D myocontrol task for each participant, using the four muscles with the highest TP that also spanned the 2-D space. Row C) Forces associated with the muscle synergies [shown on Fig 3]; Row D) The reduced mapping for the Select Synergy condition in the 2-D myocontrol task, using the four synergies with the highest TP that also spanned the 2-D space.

For each participant the number of synergies adequately capturing the EMG data (*N*) was selected according to the fraction of data variation explained, defined as

$$R^2_{EMG} = 1 - \frac{SSE_{EMG}}{SST_{EMG}} \tag{4}$$

where $SSE_{EMG}$ is the sum of the squared muscle activation residuals and $SST_{EMG}$ is the sum of the squared residuals of the muscle activation from its mean vector. We extracted a minimum of five synergies and additionally considered two criteria. The first criterion was a threshold of 0.9 on $R^2_{EMG}$. The second EMG criterion was based on the detection of a "knee" in the slope in the curve of the $R^2$ value as a function of *N*. A series of linear regressions was performed on the portions of the curve included between *N* and its last point (*M*). N was then selected as the minimum value for which the mean squared error of the linear regression was less than $10^{-4}$ [27,38]. In case of mismatch between the two criteria, the larger *N* was chosen. Extracted synergies for each participant are shown in Fig 3.

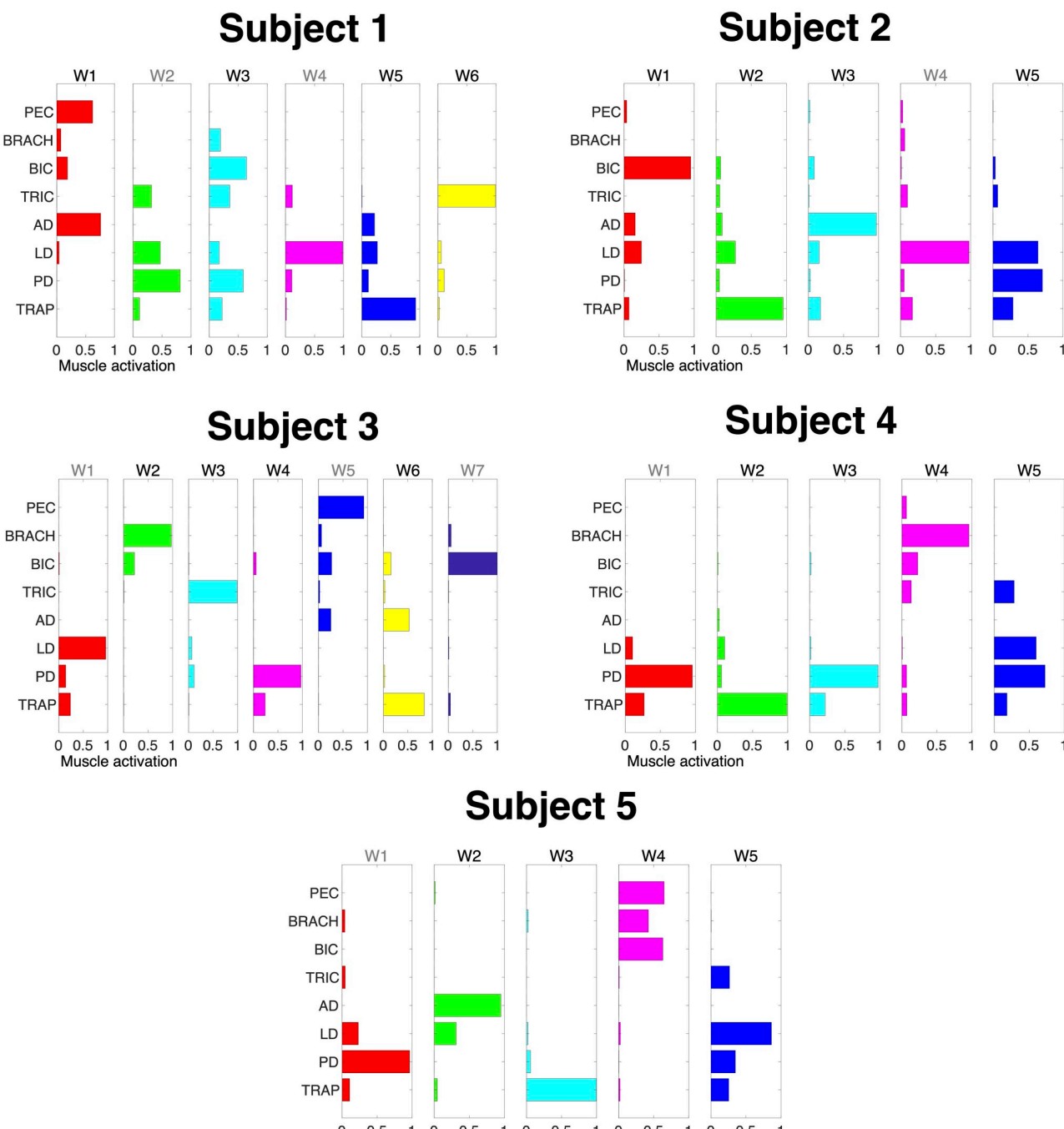

**Fig 3. Muscle synergies.** Muscle synergies (matrix W) identified by non-negative matrix factorization from the EMG data of each participant (columns) collected in the force-control block. Each column of W is a vector specifying a specific pattern of relative level of muscle activation. Synergies in black represent the ones selected for use in the 2-D myocontrol task for the Select Synergy condition. Pectoralis major (PEC), brachioradialis (BRACH), biceps (BIC), triceps (TRIC), anterior deltoid (AD), lateral deltoid (LD), posterior deltoid (PD), and middle trapezius (TRAP). W$n$ in black represent the synergies selected after assessment with the 1-D task.

## Muscle and synergy selection

To select the muscles and synergies to be used in the 2-D myocontrol task, participants performed a 1-D isometric speed-accuracy task. Specifically, participants were instructed to move a cursor from rest position into targets of different sizes as fast and accurately as possible. Prior to each target, subjects were asked to relax and keep the cursor at the bottom of the screen. As soon as the target appeared, subjects were instructed to reach the target as quickly as possible and maintain the position for 500 ms. The movement was constrained to one axis in the pulling direction for either each muscle and synergy. This allowed to constrain the movement in only one direction and to assess each muscle and synergy individually.

Vertical cursor position was linearly proportional to either the synergy or muscle activation such that 0% MVF was the bottom of the screen and 50% MVF was the top of the screen. For each synergy or muscle, participants completed 40 trials, reaching to four targets of different widths (3.0, 4.3, 6.1, 8.6% MVF) and distances (7.5 and 15% MVF) in each component's natural pulling direction calculated from the EMG-to-Force mapping (Fig 2, Row A and Row C). IDs for the 1-D task ranged from 0.80 to 3.32 bits.

Muscles and synergies were then ranked based on average information throughput (TP), which is defined as the quantitative measure of controllability in terms of bit rate [32]:

$$\overline{TP} = \frac{1}{x} \sum\nolimits_{i=1}^{x} \frac{ID_i}{MT_i} \tag{5}$$

where $x$ is the number of targets, and MT is the movement time to reach the target.

The dimensionality of the muscle and synergy mappings was then reduced by eliminating less controllable muscles or synergies. The 4 muscles and synergies with the highest TP were selected for the 2-D task. The criterion for selecting the synergies was based on a previous study showing that in healthy subjects the trajectories reconstructed using 4 synergies were as accurate as the trajectories estimated from EMGs recorded from the entire set of muscles [27]. If the pulling direction of a muscle or synergy made an angle of 60° or less with that of a muscle or synergy with higher throughput, then we did not select it in order to ensure that the selected muscles and synergies spanned the 2-D space (Fig 2, Row B and Row D). The angular difference between two muscles or synergies was calculated as the inverse cosine of their pulling directions.

## EMG- and synergy-control

As proposed in a previous study [27], output forces f during EMG-control were computed using the EMG-to-force mapping (H) and the recorded muscle activity m by:

$$f = Hm \tag{6}$$

thus, allowing for individual muscle control. During synergy-control, muscle activity was substituted by the product of the initially extracted subject-specific synergies (W) and estimated synergy coefficients ($\hat{c}$), i.e., by f = HW$\hat{c}$, where HW is the synergy-to-force mapping, and each column of W is the pulling direction of each synergy. Synergy coefficients were estimated by projecting recorded muscle activity onto the synergy space, i.e., by $\hat{c}$ = W$^+$m, where W$^+$ is the pseudo-inverse of W, corresponding to estimating $\hat{c}$ from m as least squares solution of m = Wc. Thus, during synergy-control, output forces were computed as:

$$f = HWW^+m \tag{7}$$

For the Select Muscle and Select Synergy control, the force mappings were reduced to only use columns corresponding to the select muscles or synergies in the force estimation. For the Select Muscle mapping, each column vector was normalized ($v/||v||$) so that EMG values were projected onto their respective axes. Without normalization, participants were unable to reach targets because the maximum force was reduced due to the absence of the excluded muscles.

## Data analysis

Data analysis was executed with Matlab R2016a (Mathworks, Natick, MA). Statistical analysis was performed using RStudio, version 0.99.903 (RStudio Inc., Boston, MA), the R-package lme4, version 1.1–12, and multcomp, version 1.4–8.

We used MT, success rate, and throughput (TP) during the speed-accuracy task as outcome measures. TP for each trial was calculated as the ratio between ID and MT, similar to its calculation in the 1-D case for muscle and synergy selection. However, in the 2-D case, the distance used to calculate ID was defined as the distance from the cursor position at target presentation to the target center. This accounts for some trials where participants were not at the origin during target presentation (± 2% MVF).

In order to compare success rate, we computed the odds ratio for the statistical analysis rather than comparing an average success over N trials. This resulted with values 0 or 1 depending on whether participants missed or hit the target. With odds ratio higher than 1 success is more likely than failure; less than 1 failure is more likely than success; and equal to 1 success and failure are equally likely.

Since the design of our experiment had interdependent measures, we used linear mixed effects analysis to express the relationships in our data instead of linear regression analysis, which requires measurements to be independent. Since success was a binomial measure, we used a generalized linear model with a logistic regression, however, the model was very similar to that used for MT and TP. All analyses included *ID* and myocontrol condition (*Condition*) as fixed effects, and intercepts for participant as a random effect, resulting in the following R regression model:

$$Dependent variable \sim ID + Condtion + (1|Subject) \tag{8}$$

Once the models were created, we compared the model including all the factors (Full) against a reduced model without the effect in question (Null) in order to test if the fixed effects significantly affected the dependent variable. In order to test interaction between the 2 fixed effects (*ID* and *Condition*), we compared the model that takes into account the interaction between fixed effects (Full) against the model without the interaction (Null). For all comparisons, p-values and Akaike's information criterion values (AIC) were obtained by likelihood ratio tests of the Full model with the Null model [40]. If the factor in question significantly affects the dependent variable, then the comparison will report a significant p-value ($< .05$) and an AIC value lower for the Full model. Similarly, a significant interaction between factors will result in a significant difference between the Full and the Null models ($p < 0.05$) with a lower AIC for the Full model. Post-hoc multiple comparisons were performed with Bonferroni correction of p-values to determine differences between specific myocontrol conditions. Differences in performance measures within individual participants were assessed either with two-sided t-test if the data were distributed normally (according to a Lilliefors test) or by Wilcoxon rank-sum with Bonferroni correction otherwise.

## Results

The likelihood ratio test showed that *ID* had a significant effect on MT (AICFull-AICNull = -86.9; $\chi^2$ = 92.88, p < 0.001) meaning the task effectively imposed the predicted speed-accuracy tradeoff in the 2-D task. On average MT was 2.44 ± 0.217 SEM, 3.19 ± 0.223 SEM and 3.56 ± 0.233 SEM s for ID 1.3, 2.05 and 2.8 respectively. MT increased with ID by 0.75 ± 0.20 seconds per bit. No significance effect was found on MT for the *Condition* fixed effect. On average MT was 3.27 ± 0.267 SEM, 2.89 ± 0.274 SEM and 3.03 ± 0.269 SEM s for All EMG, Select Muscle and Select Synergy respectively. Analysis of individual participants showed that MT for Select Muscle was significantly lower than MT in Select Synergy only for participant 2 (p < 0.001). While, participant 5 showed higher MT for Select Muscle compared to All EMG and Select Synergy. Results for MT are shown in Fig 4.

The likelihood ratio test showed that ID had a significant effect on success rate (AICFull-AICNull = -98; p < 0.0001). The odds ratio associated with an increase in ID was on average 0.44 across all subjects, namely the success rate decreased with increased of ID. Condition also had a significant effect on success rate (AICFull-AICNull = -27; p < 0.0001). Post-hoc pairwise comparison showed that there was a significant difference between success rate in Select Muscle and All EMG, with Select Muscle having an average odds ratio of 0.44 compared to All EMG (p < 0.0001), signify a decrease of success rate with Select Muscle compared to All EMG. There was also a significant decreased of success rate with Select Synergy relative to All EMG,

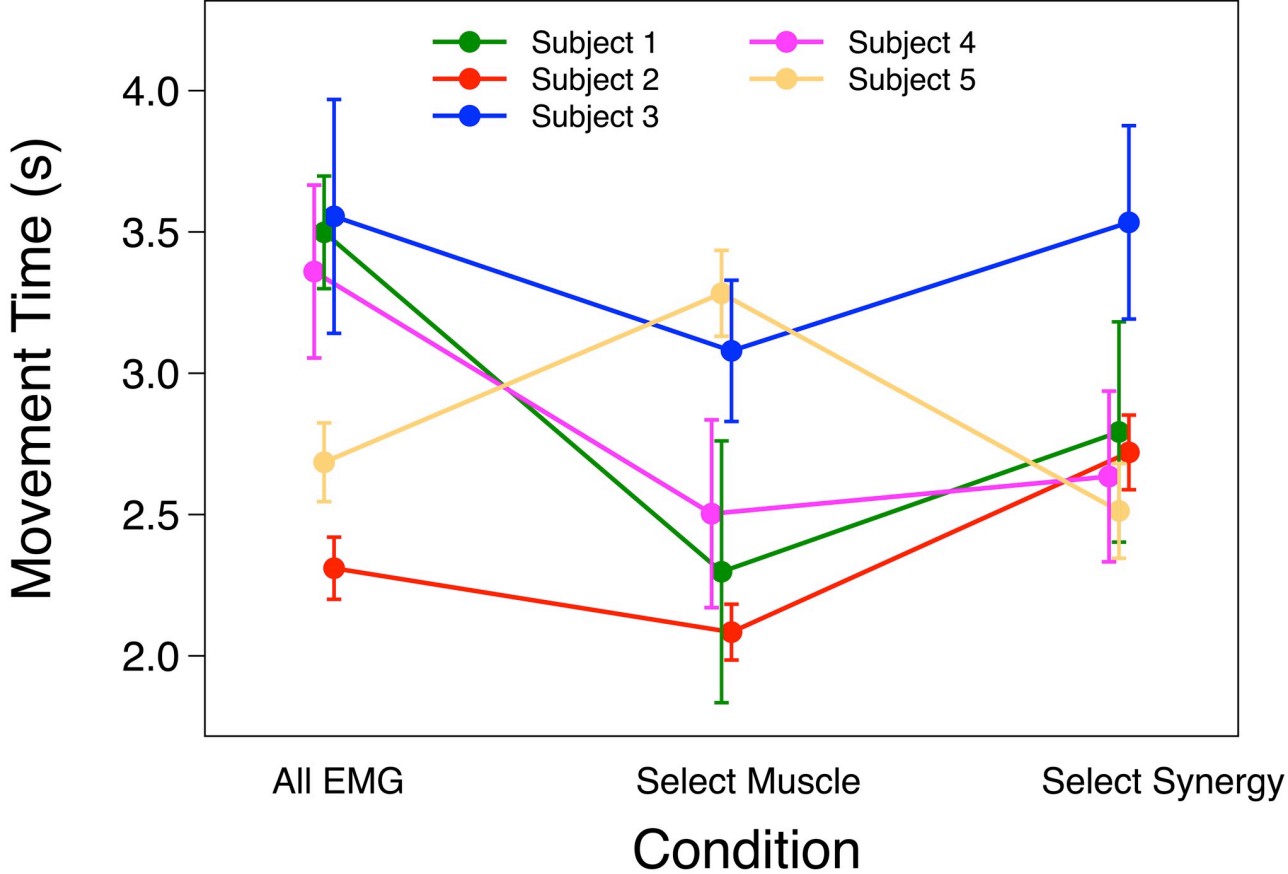

**Fig 4. Movement time.** Mean and SEM of movement time (MT) of each participant using All EMG, Select Muscle, and Select Synergy control.

**Table 2. Success rate of individual subjects for each condition.**

| Subject | All EMG | Select Muscle | Select Synergy |
|---------|---------|---------------|----------------|
| 1 | 0.32 | 0.07 | 0.11 |
| 2 | 0.78 | 0.72 | 0.64 |
| 3 | 0.1 | 0.2 | 0.18 |
| 4 | 0.24 | 0.17 | 0.21 |
| 5 | 0.5 | 0.53 | 0.45 |

with Select Synergy having an average odds ratio of 0.30 compared to All EMG (p < 0.0001). There was no significant difference between Select Synergy and Select Muscle (p = 0.94, odds ratio = 1.07). Additionally, interaction between Condition and ID had a significant effect on success rate (AICFull-AICNull = -13; p < 0.001), meaning that change in success rate across ID differed across Conditions. The proportion of successful trials for each participant is listed in Table 2.

The likelihood ratio test showed that ID and Condition fixed effects were not significant on TP. On average TP was 0.909 ± 0.101 SEM, 1.087 ± 0.106 SEM and 1.073 ± 0.103 SEM bits/s for All EMG, Select Muscle and Select Synergy respectively. Individual analysis for TP showed a significance only for participant 5, such that TP was greater Select Synergy compared to Select Muscle. TP results are shown in Fig 5.

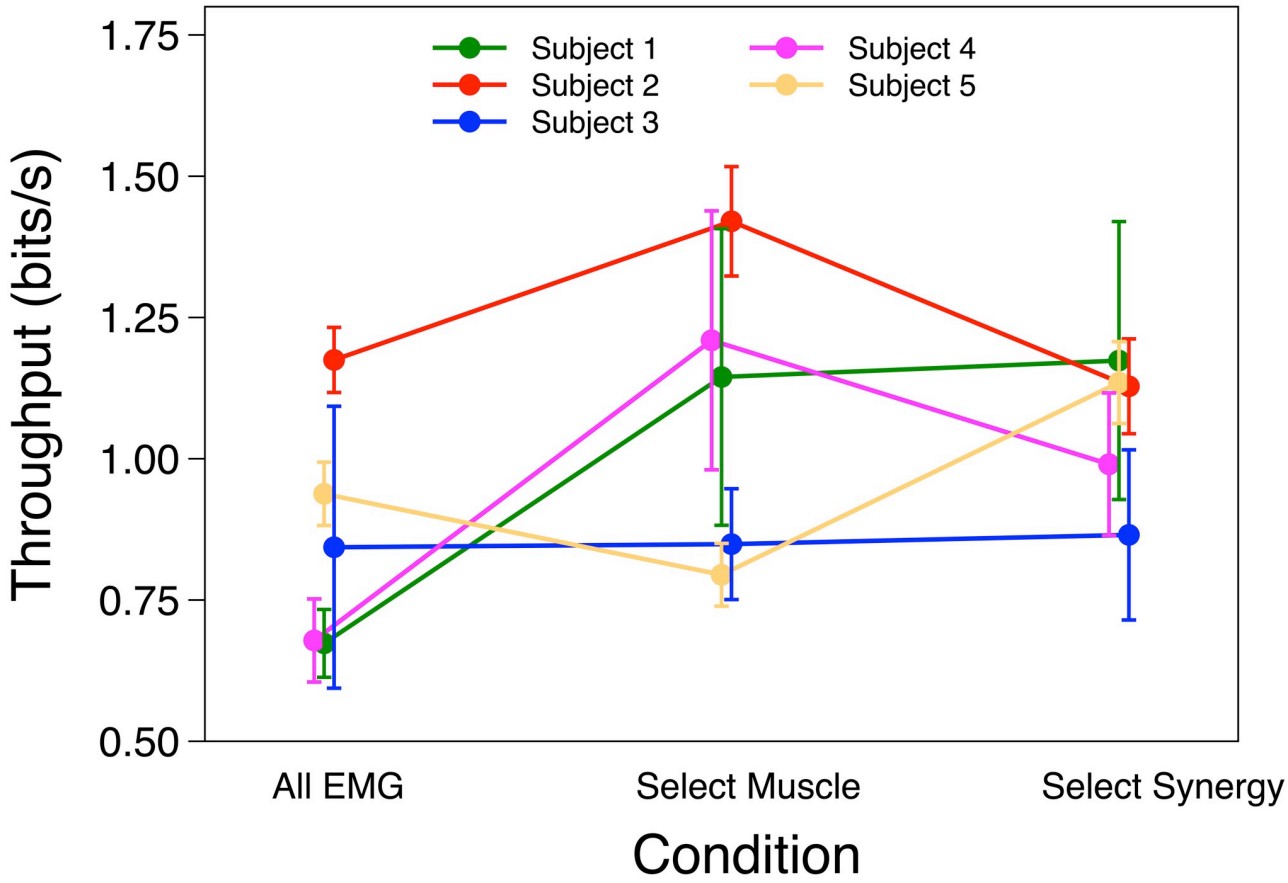

**Fig 5. Throughput.** Mean and SEM of Throughput (TP) of each participant using All EMG, Select Muscle and Select Synergy control.

## Discussion

We wanted to see if the removal of less controllable synergies or muscles could improve myocontrol performance compared to All EMG myocontrol in children with dyskinetic cerebral palsy (CP). The experimental task was designed using the Fitts' Law paradigm to test the controllability of components identified by synergy decomposition and selected muscles. A linear relationship was found between the MT to achieve the target and the ID, as expressed by the Fitts' model, for all the experimental conditions (i.e. All Muscle, Select Muscle and Select Synergy). The findings are in accordance with previous studies reporting that myocontrol performances followed the Fitts' Law in healthy control subjects [33,34,37] and patients with dyskinetic CP and primary dystonia [41]. However, the overall results did not show significant changes in performance for Select Muscle and Select Synergy compared to All Muscle condition either for MT, TP and success rate. Based on these outcome measures, we conclude that removal of less-controllable components does not adequately improve myocontrol performance. It appears that children develop a strategy that uses all available muscles to improve performance, so that removal of the contribution of any muscle or combination of muscles reduces the available degrees of freedom and reduces performance.

Based on our results, we can infer that controllable and involuntary components are present on all EMG channels. The involuntary components cannot be isolated to individual muscles or to low-dimensional synergistic muscle patterns. Previous literature suggests that synergies in children with dyskinetic CP may be similar to those in typically developing children. In a study comparing reaching and "Figure 8" drawing movements, children with dystonia and typically developing children shared many similar synergies, however time activations differed [42]. It was found that both children with CP and typically developing children had similar structure of motor output in gait, but children with CP exhibited wider temporal activation patterns [43]. Our findings further confirm the results reported in previous studies suggesting that dystonia in dyskinetic CP is characterized by an impaired ability to properly suppress variable and task uncorrelated muscle activity [12,44]. Both primary and secondary dystonia have been associated with injury or dysfunction in the basal ganglia. It has been suggested that abnormalities arising from basal ganglia malfunction in dystonia may be related to the inability to remove unwanted components of movement [45,46]. Therefore, motor impairments in dyskinetic CP could represent the superimposition of unwanted muscle activations on the desired movement pattern, rather than additional or differing structures of motor modules during the motor execution [42,47]. Moreover, it has been suggested that hyperkinetic and unwanted muscle activations in upper extremities in children with dyskinetic CP are characterized by increased signal-dependent noise of the motor system [13–15,24]. This means that children dyskinetic CP in our study would have also required significantly slower movements to contain the increased motor variability and resulting in poorer controllability of myocontrol [41].

There are some studies showing that children with CP have different synergies than typically developing children. Some have observed that children with CP recruit fewer synergies in walking [48,49]. A previous study observed CP-specific synergies [49], however, these results were only tested in lower extremities. Discrepancies among these studies may be due to different numbers of muscles and different criteria for synergy selection [43,48–50]. While removing less controllable synergies does not improve myocontrol performance for children with dyskinetic CP, a different approach may involve using synergies to aid learning rather than execution. A study done on adults showed different adaptation rates in a myocontrol task depending on the span of the synergies [38]. For children with dyskinetic CP, projecting to a

synergy space different from their own may filter noise from some dimensions not necessary for control while allowing participants to learn to control a lower dimensional space. Future work would be needed to determine whether the exploitation of learning environment by removing or manipulating specific synergies components or muscles could promote functional synergy recruitment and synergy reorganization resulting with a better myocontrol performance.

Our study has some limitations that will need to be addressed in future research. A larger cohort would be needed to derive final conclusions on the effects of filtering to extract the relevant components to improve myocontrol interfaces in children with dyskinetic CP. Subject 2 was characterized by mild symptoms of dystonia (i.e. low rating of BAD scale), and performed the myocontrol task with the highest success rate compared to the other subjects. This suggests that the exacerbation of motor impairments increases the extent to which the unwanted components muscle activations interfere with the voluntary components during myoelectrical activity. Therefore, it would be worthwhile to further investigate how severity of motor impairments in dyskinetic CP effects the filtering of controllable components on EMG control performance.

The H matrices and force pulling directions of muscles were inconsistent among the participants. This could be caused by involuntary sustained muscle contractions and co-activations between agonists and antagonists, which may result in in different pulling force direction respect to the expected ones. However, even if the pulling directions were not estimated very accurately, they were used for both muscle and synergy control, and therefore marginally affect the comparison between control conditions.

This study sought to determine if spatially filtering EMG from children with dyskinetic CP would improve myocontrol performance. We have learned that selecting the most controllable muscles or synergies as quantified by Fitts' Law does not improve myocontrol performance for children with dyskinetic CP, and can make performance even worse in some subjects (e.g. higher success rate for Subject 1 with the All Muscle condition). This appears to result from the spread of controllability and unwanted muscle activations over all available dimensions of control. As a result, the involuntary movement components are not confined to a low-dimensional subspace, and they cannot be removed by subspace filtering. Thus, in order to establish myocontrol for children with dyskinetic CP, future work should be focused on temporally filtering muscle or synergy activations, rather than spatially filtering the less controllable muscles or synergies. Further, it suggests that myocontrol in children with dyskinetic CP may benefit from a larger number of muscles, even with increased uncontrollable neuromotor noise [14,22,24]. This work provides insight in where inherent motor variability is distributed across the set of controllable synergies and is not confined to a single low-dimensional pattern. As we develop a better understanding on the mechanism of impaired ability to appropriately suppress unwanted and irrelevant muscle activity in dyskinetic CP, we may be able to develop a nonlinear myocontrol filter to assist in removal of such noise, allowing children with dyskinetic CP to control assistive devices to perform movements that they are otherwise unable to perform with their own limbs.

## Supporting information

**S1 Fig. Data source of the figures.**
(ZIP)

**S1 Data. Performance data.**
(XLSX)

## Acknowledgments

The authors would like to thank Aprille Tongol and Diana Ferman for assistance with participants recruitment.

## Author Contributions

**Conceptualization:** Cassie N. Borish, Andrea d'Avella, Terence D. Sanger.

**Data curation:** Cassie N. Borish, Matteo Bertucco.

**Formal analysis:** Cassie N. Borish, Matteo Bertucco.

**Funding acquisition:** Terence D. Sanger.

**Investigation:** Cassie N. Borish.

**Methodology:** Cassie N. Borish, Matteo Bertucco, Denise J. Berger, Andrea d'Avella, Terence D. Sanger.

**Software:** Cassie N. Borish.

**Supervision:** Matteo Bertucco, Terence D. Sanger.

**Writing – original draft:** Cassie N. Borish, Matteo Bertucco.

**Writing – review & editing:** Denise J. Berger, Andrea d'Avella, Terence D. Sanger.

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
