## [Decision Letter · Decision Letter 0]

16 Dec 2020

PONE-D-20-20429

Can spatial filtering separate voluntary and involuntary components in children with dyskinetic cerebral palsy?

PLOS ONE

Dear Dr. Bertucco,

Thank you for submitting your manuscript to PLOS ONE. After careful consideration, we feel that it has merit but does not fully meet PLOS ONE’s publication criteria as it currently stands. Therefore, we invite you to submit a revised version of the manuscript that addresses the points raised during the review process.

Both reviewers are in agreement about the value of this submission and I share their interest in the study. The review identified significant concerns about study design and the clarity of presented information. Adding the representative examples that walk the reader through the general analysis and results would improve the clarity of methodological description. The description of limitations should be expanded, and alternative explanations could be discussed. 

We look forward to receiving your revised manuscript.

Kind regards,

Sergiy Yakovenko

Academic Editor

PLOS ONE

Journal Requirements:

https://doi.org/10.1016/j.ijhcs.2020.102432. Specifically the paragraph in the introduction of your submission which begins: 'Fitts’ Law is the well-known mathematical formulation....'

In your revision please rephrase any duplicated text - further consideration is dependent on these concerns being addressed.

4. Please upload a copy of Figure 8, to which you refer in your text. If the figure is no longer to be included as part of the submission please remove all reference to it within the text.

Reviewers' comments:

Reviewer's Responses to Questions

**Comments to the Author**

1. Is the manuscript technically sound, and do the data support the conclusions?

Reviewer #1: Partly

Reviewer #2: Partly

2. Has the statistical analysis been performed appropriately and rigorously? 

Reviewer #1: I Don't Know

Reviewer #2: Yes

3. Have the authors made all data underlying the findings in their manuscript fully available?

Reviewer #1: Yes

Reviewer #2: Yes

4. Is the manuscript presented in an intelligible fashion and written in standard English?

Reviewer #1: Yes

Reviewer #2: Yes

5. Review Comments to the Author

Reviewer #1: This is a highly relevant, well conducted and articulated research.

There is concern regarding the multi-phase nature of this device control study, that extends the application of results to reveal movement mechanisms underlying dyskinetic CP, leading to over-interpretation of results. This is a complex study in a very small sample and these limitations need to be recognized. There are many potentially confounding variables at each phase of the study i.e., other explanations, that would change implications for the movement mechanisms underlying dyskinetic CP. The implications for device myo-control are appropriately discussed.

To determine if identifiable dyskinetic movement patterns exist and imposed on/within voluntary movements in dyskinetic CP, movement profiles and EMG patterns can be and are analyzed in detail. These results have direct implications on movement mechanism underlying dyskinetic CP and should be discussed.

Further phases of the study i.e., elimination of muscles/ EMG components and use of the device myo-control, introduce too many confounds to have meaningful implications for movement mechanism. These results do have direct implications for device control strategy and myo-control performance. I recommend eliminating the sentence in the Abstract and on p.27 regarding, ‘This work provides insight in where the noise lies…’ and contain comments on movement mechanism to results of the first phase of the study.

Reviewer #2: The work reports clearly describes an interesting study. However, some clarifications and further details are needed.

Experimental protocol

• Were the 2 force-control blocks carried out before each of the 3 conditions?

• It is not clear the goal of the first force-control block. Is to estimate subject-specific MVF? And then to compute MT? Therefore, wasn’t MT computed during the 240 trials for each condition?

• Was the second force-control block used to compute H and ef (from equation 2)? It might be explicitly linked in the text

• MT was set at 7.5 seconds as maximum, how was this value defined?

• “widths 4.31, 7.24, and 12.18 % MVF, resulting in IDs equal to 1.3, 2.05, and 2.8 bits respectively..” Shouldn’t ID and W be inversely “proportional”?

• Why is there a different range between force-control block and “Select Muscle” and “Select Synergy” condition blocks? 0-50% MVC and 0-25% MVC

• Why was the movement onset in some cases considered as 5% of peak velocity and in other cases the time when the target appears?

Data analysis

Not very clear the paragraph about the success rate. Later in results, in Table 2, the values are from 0 to 1, as proportion of successful trials. How are they linked?

Results

From Fig 2

• Isn’t there any repeatability of planar force amplitude neither of pull direction generated by each muscle (row A), across the 5 subjects?

• To clarify the muscle selection: “the 4 muscles with highest TP that also spanned the 2-D space”. They are not the most involved in row A (e.g. m 2 of subj. 5 in row A bigger than muscle 7, but muscle 7 is selected in row B)

• Row B: Subj 3 changed colour (e.g. muscle 4 from green in row A to light blue in row B), what does it mean?

• The number of synergies is free? 5 6 or 7 synergies for the different subjects

• How the 4 synergies are selected? always 4? Aren’t they the ones producing higher force (e.g. subj 3, synergy 7 (mainly BIC) is bigger than synergy 4 (mainly PD), but synergy 4 is selected)

• The two synergies cited above are even quite parallel; do BIC and PD produce similar planar forces?

From Fig 3

• Is there any order/hierarchy of synergies based on explained variance?

• Which is the 1-D task used to select the synergies (in black)? Why not using the 2-D task itself?

About fig 4:

• It seems that participant 2 shows MT for Select Muscle also lower than MT in All EMG, isn’t it?

• Subjects 1 and 4 seem to have “significant” higher MT in All EMG, isn’t it?

About odds ratio.

• When reporting 0.44, or 0.30 in the text, are they averaged across subjects?

• Table 2. Subject 1 in “All EMG” has a very higher success rate than in the other 2 conditions. But this could be due to a slower movement in All EMG. It might be discussed

• Is there any correlation between severity and outcome measure? E.g. the less severe subject (subj 2 with BAD 1 on tested arm) seems to have highest success rate

About TP.

• SEM: how is it defined? (in Fig 5 Standard Error (SE) is reported and cited in caption)

• Subjects 1 and 4 seem to have “significant” higher TP in Select Muscle than in All EMG, isn’t it?

Discussion

• The first claim “We found that Select Synergy and Select Muscle myocontrol had higher movement time and lower success than All EMG myocontrol” doesn’t seem to be strongly supported by data

• It may be better explained how a different synergy space could improve control through a more effective learning

• The main conclusion is that the EMGs of CP children are overall corrupted by noise. It should be better discussed if it was already a proposed theory in previous works

Few typos, e.g.

o “postion”,

o TP acronym used before definition,

o Fig 2 caption: reference to Fig2 itself, I think it should be Fig3

o ..

6. PLOS authors have the option to publish the peer review history of their article (what does this mean?). If published, this will include your full peer review and any attached files.

Reviewer #1: No

Reviewer #2: No

---

## [Author Response · Author response to Decision Letter 0]

22 Jan 2021

We are very grateful to the anonymous Reviewers and the Editor for the many productive comments and suggestions. In the revised manuscript, we believe we have addressed all their concerns.

We uploaded the responses to the Reviewers and to the Editor as a word file (.doc) in the Attach Files

---

## [Decision Letter · Decision Letter 1]

30 Mar 2021

Can spatial filtering separate voluntary and involuntary components in children with dyskinetic cerebral palsy?

PONE-D-20-20429R1

Dear Dr. Bertucco,

We’re pleased to inform you that your manuscript has been judged scientifically suitable for publication and will be formally accepted for publication once it meets all outstanding technical requirements.

Kind regards,

Sergiy Yakovenko

Academic Editor

PLOS ONE

Additional Editor Comments (optional):

Thank you for the thorough review and apologies for the delay. I had trouble recruiting one of the original reviewers; however, I believe the concerns have been adequately addressed. My own review has not identified any additional obstacles for the publication of this manuscript.

Reviewers' comments:

Reviewer's Responses to Questions

**Comments to the Author**

1. If the authors have adequately addressed your comments raised in a previous round of review and you feel that this manuscript is now acceptable for publication, you may indicate that here to bypass the “Comments to the Author” section, enter your conflict of interest statement in the “Confidential to Editor” section, and submit your "Accept" recommendation.

Reviewer #2: All comments have been addressed

2. Is the manuscript technically sound, and do the data support the conclusions?

Reviewer #2: Yes

3. Has the statistical analysis been performed appropriately and rigorously? 

Reviewer #2: Yes

4. Have the authors made all data underlying the findings in their manuscript fully available?

Reviewer #2: Yes

5. Is the manuscript presented in an intelligible fashion and written in standard English?

Reviewer #2: Yes

6. Review Comments to the Author

Reviewer #2: (No Response)

7. PLOS authors have the option to publish the peer review history of their article (what does this mean?). If published, this will include your full peer review and any attached files.

Reviewer #2: **Yes: **Claudia Casellato

---

## [Editor Report · Acceptance letter]

1 Apr 2021

PONE-D-20-20429R1 

Can spatial filtering separate voluntary and involuntary components in children with dyskinetic cerebral palsy? 

Dear Dr. Bertucco:

I'm pleased to inform you that your manuscript has been deemed suitable for publication in PLOS ONE. Congratulations! Your manuscript is now with our production department. 

Kind regards, 

on behalf of

Dr. Sergiy Yakovenko 

Academic Editor

PLOS ONE